# Glycosylation in Indolent, Significant and Aggressive Prostate Cancer by Automated High-Throughput *N*-Glycan Profiling

**DOI:** 10.3390/ijms21239233

**Published:** 2020-12-03

**Authors:** Sarah Gilgunn, Keefe Murphy, Henning Stöckmann, Paul J. Conroy, T. Brendan Murphy, R. William Watson, Richard J. O’Kennedy, Pauline M. Rudd, Radka Saldova

**Affiliations:** 1School of Biotechnology, Dublin City University, D09 V209 Dublin 9, Ireland; sgilgunn@gmail.com (S.G.); richard.okennedy@dcu.ie (R.J.O.); 2National Centre for Sensor Research, Biomedical Diagnostics Institute, Dublin City University, D09 V209 Dublin 9, Ireland; 3Department of Mathematics and Statistics, Maynooth University, Maynooth, W23 F2K8 Co. Kildare, Ireland; keefe.murphy@mu.ie; 4NIBRT GlycoScience Group, National Institute for Bioprocessing Research and Training, Fosters Avenue, Mount Merrion, Blackrock, A94 X099 Co. Dublin, Ireland; henning.stoeckmann@gmail.com (H.S.); pauline.rudd@ucd.ie (P.M.R.); 5Department of Biochemistry and Molecular Biology, Faculty of Medicine, Nursing and Health Science, Monash University, Melbourne, VIC 3800, Australia; paul.conroy3@gmail.com; 6UCD School of Mathematics and Statistics, University College Dublin, D04 V1W8 Dublin 4, Ireland; brendan.murphy@ucd.ie; 7Conway Institute of Biomolecular and Biomedical Research, University College Dublin, D04 V1W8 Dublin 4, Ireland; william.watson@ucd.ie; 8UCD School of Medicine, College of Health and Agricultural Science, University College Dublin, D04 V1W8 Dublin 4, Ireland; 9Research, Development and Innovation, Qatar Foundation, Luqta Street, Doha 5825, Qatar; 10Bioprocessing Technology Institute, 20 Biopolis Way, #06-01 Centros, Singapore 138668, Singapore

**Keywords:** prostate cancer, biomarkers, glycosylation, indolent, aggressive, significant

## Abstract

The diagnosis and treatment of prostate cancer (PCa) is a major health-care concern worldwide. This cancer can manifest itself in many distinct forms and the transition from clinically indolent PCa to the more invasive aggressive form remains poorly understood. It is now universally accepted that glycan expression patterns change with the cellular modifications that accompany the onset of tumorigenesis. The aim of this study was to investigate if differential glycosylation patterns could distinguish between indolent, significant, and aggressive PCa. Whole serum *N*-glycan profiling was carried out on 117 prostate cancer patients’ serum using our automated, high-throughput analysis platform for glycan-profiling which utilizes ultra-performance liquid chromatography (UPLC) to obtain high resolution separation of *N*-linked glycans released from the serum glycoproteins. We observed increases in hybrid, oligomannose, and biantennary digalactosylated monosialylated glycans (M5A1G1S1, M8, and A2G2S1), bisecting glycans (A2B, A2(6)BG1) and monoantennary glycans (A1), and decreases in triantennary trigalactosylated trisialylated glycans with and without core fucose (A3G3S3 and FA3G3S3) with PCa progression from indolent through significant and aggressive disease. These changes give us an insight into the disease pathogenesis and identify potential biomarkers for monitoring the PCa progression, however these need further confirmation studies.

## 1. Introduction

Prostate cancer (PCa) is a major health-care concern worldwide. It is the second most frequently diagnosed cancer in men and fifth leading cause of death worldwide, with higher prevalence in developed countries [1]. The incidence and mortality also correlate with increasing age [1]. A total of 1,276,106 new cases of prostate cancer were reported worldwide in 2018 resulting in 358,989 deaths [1]. This cancer can manifest itself in many distinct forms from clinically indolent PCa to the more aggressive invasive form [2]. The correct stratification of patients with these forms of the disease has important implications for their treatment and can impact men worldwide socio-economically and in quality of life [3].

The modification of the vast majority of secreted eukaryotic proteins with glycan structures is one of the most common post-translational modifications. It is now universally accepted that glycan expression patterns change with changes in the cellular environment that accompany the onset of tumorigenesis [4]. The size, diversity, and complexity of glycans in mammalian systems provides a wealth of information and scientists are faced with a significant analytical challenge in understanding which glycans contribute to specific biological functions [5]. Advances in the field of proteomics and glycomics has helped glycobiologists decipher the link between glycan structures and disease progression [4].

That said, many glycan sample preparation workflows are impacted considerably by low sample throughput, lack of automation, and extremely high consumables costs, significantly hampering the urgently needed progress in glycomics research [6]. The high throughput methods devised in the GlycoScience Lab in The National Institute for Bioprocessing Research and Training (NIBRT) provide a reliable and robust method that allows for detailed analysis of femtomoles of *N*-linked sugars released from glycoproteins. This approach was designed specifically for the identification and high-throughput screening for disease biomarkers [7,8].

An automated workflow for sample preparation implemented on a robotic liquid handling platform was developed [5,6] to improve the time and efficiency of the methods previously developed by Royle and colleagues [8,9]. This system allows for the preparation of 96 samples (or multiples thereof) in 22 h with excellent reproducibility. It also improves on the HPLC technique by use of hydrophilic interaction chromatography ultra-performance liquid chromatography (HILIC-UPLC) which allows for shorter run times and greatly increased resolution [10]. HILIC glycan analysis is continuously advancing and has many advantages over other methods such as high performance anion-exchange chromatography (HPAEC) reversed-phase liquid chromatography (RP−LC) and capillary electrophoresis (CE) [10]. In HILIC, separation is based on the hydrophilicity of the glycan, which is affected by its composition, size, charge structure, linkage, and branching. The separation of structural isomers is difficult to achieve with other methodologies, which is one of main advantages of HILIC [10,11]. Saldova and co-workers showed that using HILIC-UPLC greatly improved separation of human serum *N*-glycans as compared with HPLC [10]. Glycomics data generated using this technology together with DNA methylation, transcripts and proteomics data, separated indolent, significant, and aggressive PCa with area under the curve (AUC) of 0.91% [12].

The aim of this study is to investigate if differential glycosylation patterns alone could distinguish between indolent, significant, and aggressive PCa, and which glycans contribute to the disease progression. The most common serum glycome alterations in prostate cancer are well documented and include sialylation, branching, and fucosylation [13,14,15,16]. Whole serum *N*-glycan profiling was carried out on 117 prostate cancer patients’ serum using our automated, high-throughput analysis platform for glyco-profiling which utilizes HILIC-UPLC to obtain high resolution separation of *N*-linked glycans released from the serum [6].

## 2. Results

### 2.1. Patient Cohort

A total of 117 patients, indolent (41), significant (32), and aggressive (44), were included in this study. These patients were assigned into three groups (i.e., indolent, significant, or aggressive PCa) based on the pathology of their disease after surgery, using Epstein’s criteria [17,18]. None of the patients received androgen-deprivation therapy (ADT) therapy (radical prostatectomy cohort only). The aim of this study is to predict the pathology of disease pre-operatively. A summary of the clinical data statistics for the full cohort is shown in Table 1 below.

### 2.2. HILIC-UPLC Profiling of PCa Patients’ Serum

A high-throughput platform, automated in a robotic 96-well plate format, for *N*-glycan analysis of human serum glycoproteins using HILIC-UPLC with fluorescence detection was recently described [10]. This high resolution HILIC-UPLC method allows potential biomarkers to be separated without the need for exoglycosidase digestion or pre-separation on weak anion exchange (WAX)-HPLC [10,19].

The serum *N*-glycome of all 117 PCa patients was analyzed in duplicates. The profiles were separated into 50 peaks and structural assignment carried out according to Saldova et al., 2014 [10]. Each peak was examined individually to see if there was any significantly different relative expression between indolent, significant, and aggressive PCa patients (Figure 1). The relative amounts of total core-, outer-arm fucose, and oligomannose glycans were calculated based on composition of these 50 peaks, and the glycans were then summarized into five groups according to sialylation (S0−S4), five groups according to galactosylation (G0−G4), and four groups according to branching (A1−A4) (see glycan features in Table 2). All the individual glycan data for each patient as well as clinical characteristics are in Appendix A.

### 2.3. Significant Changes in Whole Serum N-Glycome in Prostate Cancer

Initially, all 50 peaks were examined individually to explore baseline differences in expression using one-way analysis of variance (ANOVA). Tukey’s post-hoc honestly significant difference (HSD) was performed to calculate the *p*-values and boxplots were used to illustrate the distribution of each statistically significant biomarker. From this analysis, a panel of 13 peaks were identified as significant, as outlined below.

Peak 4, Peak 7, Peak 13, Peak 15, Peak 17, Peak 18, Peak 22, Peak 27, Peak 33, Peak 34, Peak 37, Peak 38, Peak 40.

When considered individually, the 13 peaks showed statistically significantly different relative expression levels across the three patient cohorts (Figure 2 and Figure 3). Of these 13 peaks, 5 peaks (peaks 4, 7, 22, 37, and 40) showed statistically significantly different expression levels as disease progresses from indolent to aggressive (Figure 2) and these will be further discussed.

Peak 4, containing mostly biantennary bisected glycans (A2B), Peak 7, containing biantennary bisected monogalactosylated glycan (A2(6)BG1), and Peak 22 containing mostly biantennary digalactosylated monosialylated glycans (A2G2S1) and hybrid and oligomannose structures (M5A1G1S1, M8) were significantly increased as disease progresses in this study.

Notable decreases were also observed, in particular triantennary trigalactosylated trisialylated glycans with and without core-fucosylation (Peak 37-A3G3S3, Peak 40-FA3G3S3), as PCa progresses from indolent to aggressive.

When we looked at the derived features, only monoantennary glycans (A1) were significantly increased with PCa progression (Figure 2).

## 3. Discussion

Whole serum *N*-glycan profiling was carried out on 117 prostate cancer patients’ serum using a highly efficient automated, high-throughput analysis platform for glyco-profiling with HILIC-UPLC. The aim of this study was to investigate if differential glycosylation patterns could distinguish between the 3 cohorts of PCa patients.

The role of these specific glycans in cancer progression are discussed in detail below.

### 3.1. Increases in Oligomannose Glycans

Hybrid and oligomannose structures (half of Peak 22; M5A1G1S(3)1, M5A1G1S(6)1 and M8 D2, D3) increase as disease progress. *N*-glycans with oligomannose, hybrid, or complex type sugar chains contribute to many cellular processes including cell-cell/cell-matrix/receptor-ligand interaction, cell signalling/growth, and differentiation [20]. Given that oligomannose structures participate both in cell survival and cell death, this could be a basis for the observed increased abundance of oligomannose structures in significant and aggressive disease compared to indolent disease.

Oligomannose *N*-glycans are increased in high-grade prostate tumors and linked to clinical outcome [16] and hybrid *N*-glycans were also increased in castration-resistant prostate cancer [21]. An abundance of oligomannose structures were observed when the *N*-glycan profiles of membrane proteins across 3 different colorectal cancer cell lines (LIM1215, LIM1899, and LIM2405) were investigated [22]. The authors of this study suggest incomplete *N*-glycan processing led to the accumulation of oligomannose type structures in the colorectal cancer cell lines. In another recent colorectal cancer study the authors showed oligomannose *N*-glycan structures were more common in carcinomas than in adenomas [23].

### 3.2. Alterations in Branching and Bisecting Glycans

In this study we observed a significant decrease in triantennary trisialylated glycans in peak 37 (A3G3S3) and peak 40 (FA3G3S3) with aggressive PCa. Biantennary digalactosylated monosialylated glycans (A2G2S1, half of Peak 22) as well as bisecting glycans in peaks 4 and 7 (A2B and A2(6)BG1) also significantly increase. When we pooled the glycans into the features we see significant increase in monoantennary glycans (A1) with disease progression.

We have noted alterations in certain glycan structures in a previous prostate cancer study that correlate with glycans in our study [13], in which significant decreases in triantennary trigalactosylated trisialylated glycans (A3G3S3) were observed in advanced aggressive PCa [13]. Patients with a Gleason score of 7 are generally positive for perineural invasion (PNI), found to be at a more advanced stage of metastasis, and have an increased chance of re-currence. PNI is an important pre-operative indicator of the pathological stage of the tumor. In our previous study we showed that decreases in triantennary trigalactosylated glycans correlate with PNI and can help diagnose tumor spread [13]. In the same previous study, decreases in A3G3S3 could distinguish between patients with Gleason score 5 and Gleason score 7 significantly better than the currently used PSA assay [13]. In this body of work, significant decreases in this structure were observed in the significant and aggressive patient groups compared to the indolent group.

Decreases in larger branched glycans as well as increases in bisecting glycans were found in prostatic secretions and correlated with disease severity [24] in line with this study’s results on serum samples from prostate cancer patients. This is contrary to the general observation in cancer, including prostate, where these trends are mostly opposite [11,14,21,24,25,26]. Branching is related to bisecting as the presence of bisecting-GlcNAc residues alter the structural conformation of the glycan chains and tend to limit branching [24]. There may be several factors involved in this observation, such as increases in branching which are observed in cancerous vs. non-cancerous samples. We have compared all cancer patients here, and there may be change in the trend once the cancer starts developing. High heterogeneity of prostate disease, the number of samples in the previous studies as well as type of samples (serum or tissues or fluid) and technique used for these evaluations also plays a role in these results. Nyalwidhe et al. found a decrease in branching and increase in bisecting glycans in prostatic fluid with prostate cancer progression [24]. Totten et al. had small number of serum samples from patients with PCa (10) and benign prostate hyperplasia (BPH) (7) [14], and Lange et al. profiled mouse model and prostate cancer cell lines for expression of branching enzyme, MGAT5b, β1,6-*N*-acetylglucosaminyltransferase-5b [25]. Kyselova et al., using Matrix-assisted laser desorption/ionization (MALDI), found a decrease in the relative intensities of oligomannose and complex biantennary structures and the concomitant increase in the fucosylated complex biantennary, complex tri- and tetraantennary *N*-glycans (both fucosylated and non-fucosylated) in 24 prostate cancer patients comparing to healthy control sera [26]. They also found established A2G2S1 had moderate ROC curve in separating PCa patients from healthy controls [26]. Increased tri and tetraantennary sialylated glycans were found in castration-resistant prostate cancer serum comparing to group of BPH, newly diagnosed PCa, and PCa patients treated with androgen-deprivation therapy without disease progression [21,27]. It should be noted here that in general, medication can affect the serum *N*-glycome [28]. Additionally, serum and tissue *N*-glycans differ as the serum glycome is typically comprised of an abundance of acute phase proteins and immunoglobulins whereas much lower levels are observed in the tissue specific glycome [29]. Hence, further validation studies with more samples are required before a conclusion on the potential clinical applications of these findings can be drawn.

## 4. Materials and Methods

### 4.1. Serum Samples

Samples were collected with consent from prostate cancer patients following a standard operating procedure, which is part of the Prostate Cancer Research Consortium (PCRC) BioResource. Ethical consent was granted from respective hospital ethics committees of the consortium. Blood samples (10 mL) were collected into anticoagulant-free tubes. Samples were coded and transported on ice to the laboratory. The tubes were centrifuged at 2500 rpm at 20 °C for 10 min within a 30-min time frame. Serum from each patient sample was then collected, aliquoted, and stored at −80 °C until time of analysis. A total of 117 patients, indolent (41), significant (32), and aggressive (44), were included in this study. Epstein’s criteria based on the final pathology was used to define patients as having indolent, significant, or aggressive PCa [17,18]. Indolent PCa was defined as tumor volume <0.5 cm^3^, organ-confined disease, and no Gleason patterns 4 or 5. Significant disease was defined as tumor volume >0.5 cm^3^, organ-confined disease, and Gleason pattern ≥4. Aggressive PCa was defined as Gleason patterns 4 or 5 and non-organ-confined, metastatic disease.

### 4.2. N-Glycan Release and Clean up

*N*-glycans were released from the PCa serum samples (5 µL) using the high-throughput automated method previously described by Stöckmann et al. (2015) [6]. Briefly, the sample denaturation was carried out with dithiothreitol in ammonium bicarbonate, at 65 °C, alkylation with iodoacetamide, and trypsin solution was added (10 µL at 40,000 U/mL) at 40 °C, finally *N*-glycans were released from the protein backbone enzymatically via PNGase F (Prozyme peptide *N*-glycanase F, GKE-5006D, 10 µL per well, 0.5 mU in 1 M ammonium bicarbonate, pH 8.0, Agilent, Santa Clara, CA, USA). The glycans were immobilized on solid supports, and excess reagents were removed on the robotic vacuum manifold. Glycans were then released from the solid supports and labelled with the fluorophore 2-aminobenzamide (2-AB). Next, glycans were cleaned up using 96-well chemically inert filter plates (Millipore Solvinert, Burlington, MA, USA, hydrophobic polytetrafluoroethylene membrane, 0.45 µm pore size) using HyperSepDiol solid phase extraction (SPE) cartridges (ThermoScientific, Waltham, MA, USA) [5,6].

### 4.3. Hydrophilic Interaction Chromatography-Ultra-Performance Liquid Chromatography (UPLC)

2-AB derivatized *N*-glycans were separated by HILIC-UPLC with fluorescence detection on a Waters Acquity UPLC H-Class instrument consisting of a binary solvent manager, sample manager, and fluorescence detector under the control of Empower 3 chromatography workstation software (Waters, Milford, MA, USA). The HILIC separations were performed using a Waters Ethylene Bridged Hybrid (BEH) Glycan column, 150 × 2.1 mm i.d., 1.7 µm BEH particles, with 50 mM ammonium formate, pH 4.4, as solvent A and MeCN as solvent B. The 30 min method was used with a linear gradient of 30–47% with buffer A at 0.56 mL min flow rate for 23 min followed by 47–70% A for 1 min and finally reverting back to 30% A in another 1 min and to complete the run [10]. An injection volume of 10 µL sample prepared in 70% *v*/*v* MeCN was used throughout. Samples were maintained at 5 °C prior to injection, while separation was carried out at 40 °C. The fluorescence detection excitation/emission wavelengths were ex = 330 nm and em = 420 nm, respectively. The system was calibrated using an external standard of hydrolyzed and 2AB-labeled glucose oligomers to create a dextran ladder, as described previously [9].

Total serum *N*-glycome from each patient was separated into 50 peaks (containing group of glycans), of individual relative proportion to total 100% peak area (relative quantitation of glycans, proportion of certain glycans comparison to total glycome).

### 4.4. Statistical Analysis

All 50 peaks and their derived features were examined individually to explore baseline differences in relative expression using one-way analysis of variance (ANOVA). Tukey’s post-hoc honestly significant difference (HSD) was performed to calculate the *p*-values. Boxplots were used to illustrate the distribution of each glycan by significance. Owing to the compositional nature of the peaks, data were pre-processed using the centered log ratio transformation prior to analysis [30]. All statistical results and Figures were obtained using the open-source R software (version 4.0.3) [31].

## 5. Conclusions

In conclusion, in this study we have observed increases in hybrid and oligomannose structures (M5A1G1S1 and M6), bisected and biantennary glycans (A2B, A2(6)BG1, and A2G2S1), and monoantennary glycans (A1), and significant decreases in triantennary trigalactosylated trisialylated glycans with and without core-fucosylation (A3G3S3 and FA3G3S3) as PCa progresses from indolent to significant to aggressive.

## Figures and Tables

**Figure 1 ijms-21-09233-f001:**
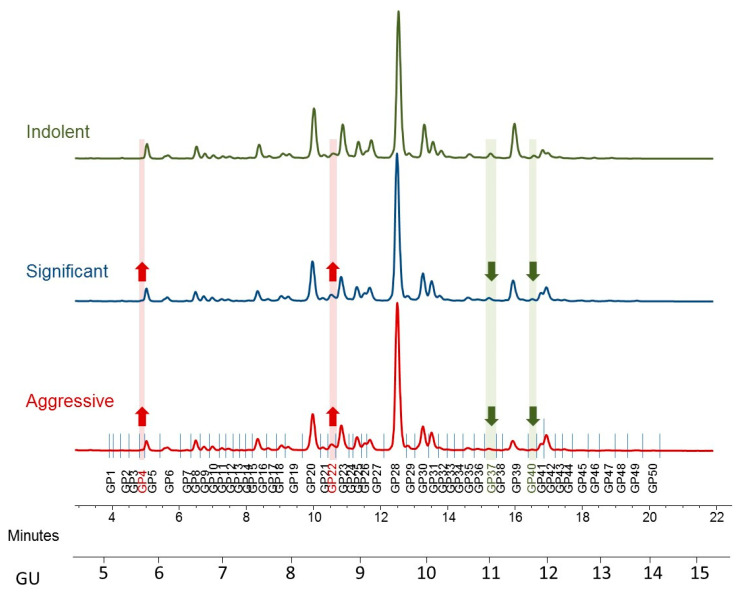
Typical hydrophilic interaction chromatography ultra-performance liquid chromatography (HILIC-UPLC) profile of undigested serum *N*-glycome from prostate cancer patients—representative profiles for indolent, significant, and aggressive disease. Highlighted are major differences: increased peaks with the disease progression are highlighted in red and decreased in green. Profiles are standardized against a dextran hydrolysate (GU, glucose unit). The HILIC-chromatogram, representing the total serum *N*-glycome from each patient, was separated into 50 peaks. Structural assignments are listed in Table 2.

**Figure 2 ijms-21-09233-f002:**
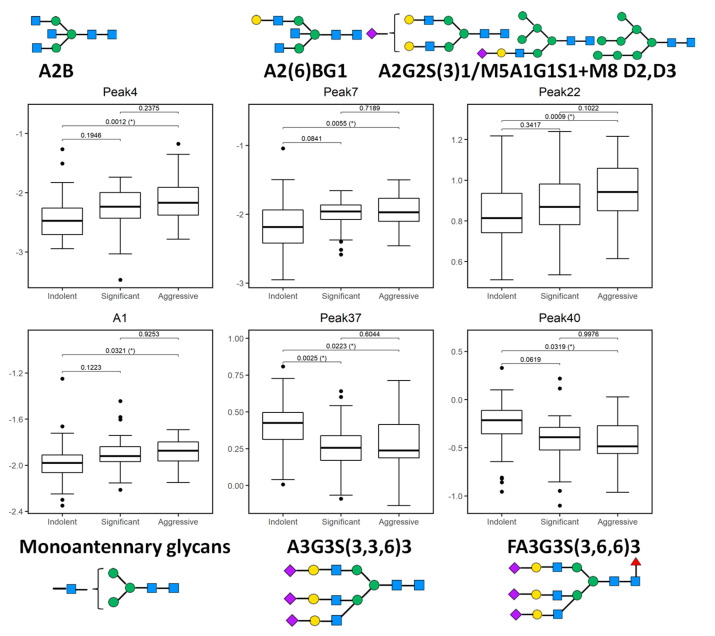
Boxplots of 5 glycosylation peaks and one feature which exhibited significant differences across 3 patient groups (indolent, significant, and aggressive) from the Irish prostate cancer research consortium (PCRC) cohort. The boxes represent the 25th and 75th percentiles with the median indicated. The lower and upper quartiles are shown as horizontal lines either side of the rectangle. The top and the first panel of the bottom row show peaks (and one feature) that significantly increase as disease progresses; otherwise the bottom row shows peaks that significantly decrease as disease progresses. From left to right, the given *p*-values relate to tests of differences between indolent and significant, indolent and aggressive, and significant and aggressive disease types, respectively, as indicated by the brackets at the top of each plot. *p*-values that are significant at the 5% level are highlighted with (*).

**Figure 3 ijms-21-09233-f003:**
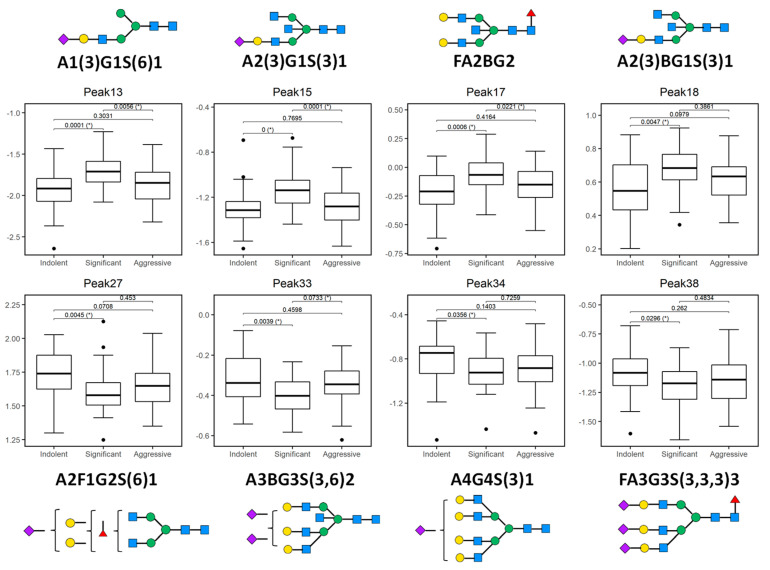
Plotted peaks from part of the PCa whole serum *N*-glycome biomarker panel—excluding those depicted in Figure 2—that show altered expression in indolent, significant, or aggressive prostate cancer (PCa) from the Irish PCRC cohort. The boxes represent the 25th and 75th percentiles with the median indicated. The lower and upper quartiles are shown as horizontal lines either side of the rectangle. From left to right, the given *p*-values relate to tests of differences between indolent and significant, indolent and aggressive, and significant and aggressive disease types, respectively, as indicated by the brackets at the top of each plot. *p*-values that are significant at the 5% level are highlighted with (*). Here, all peaks show a statistically significant difference between indolent and significant disease types, while peaks 13, 15, 17 and 33 additionally show a significant difference between significant and aggressive disease.

**Table 1 ijms-21-09233-t001:** Summary statistics for full cohort (n = 117), with breakdowns by indolent, significant, and aggressive disease type. Pre-operative radical prostatectomy PSA—Pre-operative PSA, digital rectal examination—DRE, year—Y, Gleason score—GS, Organ confined—OC, None Organ confined—NOC.

	Full Cohort (n = 117)	Indolent (n = 41)	Significant (n = 32)	Aggressive (n = 44)
		Age (y)		
Mean (median)	60.6 (60)	59.1 (59)	59.9 (59.5)	62.6 (62.5)
Range	49–74	49–70	49–73	52–74
Pre-Op PSA (ng/mL)				
Mean (median)	7.05 (6.5)	6.5 (6)	7.11 (6.3)	7.51 (6.7)
Range	0.7–18.7	0.7–12.3	2.8–14.7	3.1–18.7
Clinical Stage-DRE (%)				
T1c	61 (52)	24 (58)	19 (59)	18 (41)
T2a	21 (18)	4 (10)	6 (19)	11 (25)
Not described	35 (30)	13 (32)	7 (22)	15 (34)
Biopsy GS (%)				
≤6	57 (49)	38 (93)	11 (34)	8 (18)
3 + 4 = 7	35 (30)	3 (7)	16 (50)	16 (36)
4 + 3 = 7	13 (11)	0 (0)	4 (13)	9 (20)
≥8	11 (9)	0 (0)	1 (3)	10 (22)
Not described	1 (1)	0 (0)	0 (0)	1 (2)
Prostatectomy Stage				
OC	73 (62)	41 (100)	32 (100)	0 (0)
NOC	44 (38)	0 (0)	0 (0)	44 (100)
Prostatectomy GS (%)				
≤6	41 (35)	41 (100)	0 (0)	0 (0)
3 + 4 = 7	45 (48)	0 (0)	27 (84)	18 (41)
4 + 3 = 7	21 (18)	0 (0)	2 (6)	19 (43)
≥8	10 (9)	0 (0)	3 (10)	7 (16)

**Table 2 ijms-21-09233-t002:** Summary of identified *N*-glycans from prostate cancer serum

*N*-glycans Total			^#^ Features																
**Glycan label**	**GU**	**Glycans**	**S0**	**S1**	**S2**	**S3**	**S4**	**G0**	**G1**	**G2**	**G3**	**G4**	**A1**	**A2**	**A3**	**A4**	**High M**	**CoreF**	**Outer arm F**
GP1	4.85	**A1**																	
GP2	5.34	M4																	
	5.41	**FA1**																	
	5.41	A2																	
GP3	5.61	**A1(6)G1**																	
GP4	5.78	**A2B**																	
	5.78	A1(3)G1																	
GP5	5.88	**FA2**																	
GP6	6.18	**M5**						**50%**						**50%**			**50%**	**50%**	
	6.24	FA2B																	
	6.24	FA1(6)G1																	
	6.24	A2(6)G1																	
	6.38	FA1(3)G1																	
	6.38	A2(3)G1																	
GP7	6.55	**A2(6)BG1**																	
GP8	6.71	A2(3)BG1																	
	6.71	**FA2(6)G1**																	
	6.71	M4A1G1																	
GP9	6.84	**FA2(3)G1**																	
GP10	6.95	**FA2(6)BG1**																	
GP11	7.08	FA2(3)BG1																	
	7.08	M6 D1,D2																	
	7.08	**M6 D3**																	
GP12	7.20	**A1(3)G1S(3)1**	**50%**	**50%**				**50%**	**50%**				**50%**	**50%**					
	7.20	**A2G2**																	
GP13	7.38	**A1(3)G1S(6)1**																	
GP14	7.38	**A2BG2**																	
GP15	7.62	**A2(3)G1S(3)1**											**50%**	**50%**				**50%**	
	7.62	A2(3)G1S(6)1																	
	7.62	FA1G1S(3)1																	
	7.62	FA1G1S(6)1																	
GP16	7.62	M5A1G1																	
	7.62	**FA2G2**																	
GP17	7.76	**FA2BG2**																	
	7.76	M7 D3																	
	7.76	A2(6)BG1S(3)1																	
	7.76	A2(6)BG1S(6)1																	
GP18	7.92	**A2(3)BG1S(3)1**																	
	7.92	A2(3)BG1S(6)1																	
	7.92	FA2(6)G1S(3)1																	
	7.92	FA2(6)G1S(6)1																	
	7.92	M4A1G1S(3)1																	
	7.92	M4A1G1S(6)1																	
	7.92	M7 D1																	
GP19	8.03	FA2(3)G1S(3)1																	
	8.03	**FA2(3)G1S(6)1**																	
	8.03	FA2(6)BG1S(3)1																	
	8.03	FA2(6)BG1S(6)1																	
	8.20	FA2(3)BG1S(3)1																	
	8.20	FA2(3)BG1S(6)1																	
GP20	8.38	A2G2S(3)1																	
	8.38	**A2G2S(6)1**																	
	8.38	A3G3																	
GP21	8.53	A2BG2S(3)1																	
	8.53	**A2BG2S(6)1**																	
GP22	8.63	M5A1G1S(3)1							**50%**	**50%**			**50%**	**50%**					
	8.63	M5A1G1S(6)1																	
	8.63	FA3G3																	
	8.63	M8 D2,D3																	
	8.63	*** A2G2S(3)1**																	
GP23	8.80	FA2G2S(3)1																	
	8.80	**FA2G2S(6)1**																	
GP24	8.80	**M8 D1,D3**																	
GP25	9.02	**FA2BG2S(3)1**																	
	9.02	FA2BG2S(6)1																	
GP26	9.21	* A2G2S(3,3)2																	
	9.21	*** A2G2S(3,6)2**																	
	9.21	* FA2G2S(3,3)2																	
GP27	9.21	A2F1G2S(3)1																	
	9.21	**A2F1G2S(6)1**																	
GP28	9.43	A3G3S(3)1																	
	9.43	A3G3S(6)1																	
	9.43	M9																	
	9.62	A3BG3S(3)1																	
	9.62	A3BG3S(6)1																	
	9.62	A2G2S(3,3)2																	
	9.62	**A2G2S(3,6)2**																	
	9.62	A2G2S(6,6)2																	
GP29	9.79	FA3G3S(3)1																	
	9.79	FA3G3S(6)1																	
	9.79	FA3BG3S(3)1																	
	9.79	A2BG2S(3,3)2																	
	9.79	**A2BG2S(3,6)2**																	
	9.79	A2BG2S(6,6)2																	
GP30	10.04	A3F1G3S(3)1																	
	10.04	FA2G2S(3,3)2																	
	10.04	**FA2G2S(3,6)2**																	
	10.04	FA2G2S(6,6)2																	
GP31	10.17	FA2BG2S(3,3)2																	
	10.17	FA2BG2S(3,6)2																	
	10.17	**FA2BG2S(6,6)2**																	
	10.17	A2F1G2S(3,3)2																	
	10.17	A2F1G2S(3,6)2																	
	10.17	A2F1G2S(6,6)2																	
	10.17	M9Glc																	
GP32	10.31	A3G3S(3,3)2																	
	10.31	**A3G3S(3,6)2**																	
	10.31	A3G3S(6,6)2																	
GP33	10.43	A3BG3S(3,3)2																	
	10.43	**A3BG3S(3,6)2**																	
	10.43	A3BG3S(6,6)2																	
GP34	10.60	**A4G4S(3)1**																	
GP35	10.77	FA3G3S(3,3)2																	
	10.77	FA3G3S(3,6)2																	
	10.77	FA3G3S(6,6)2																	
	10.77	*** A3G3S(3,3)2**																	
GP36	10.96	*** A3G3S(3,3,3)3**																	
	10.96	A3F1G3S(3,3)2																	
	10.96	A4G4S(6)1																	
GP37	11.14	*** A3G3S(3,3,6)3**																	
	11.14	* A3G3S(3,6,6)3																	
	11.14	* A3BG3S(3,3,3)3																	
	11.14	* A3BG3S(3,3,6)3																	
	11.14	* A3BG3S(3,6,6)3																	
GP38	11.28	*** FA3G3S(3,3,3)3**																	
	11.28	* FA3BG3S(3,3,3)3																	
GP39	11.54	A4G4S(3,6)2																	
	11.54	A3G3S(3,3,3)3																	
	11.54	**A3G3S(3,3,6)3**																	
	11.54	A3G3S(3,6,6)3																	
	11.54	A3G3S(6,6,6)3																	
	11.72	A3BG3S(3,3,3)3																	
	11.72	A3BG3S(3,3,6)3																	
	11.72	A3BG3S(6,6,6)3																	
GP40	11.89	FA3G3S(3,3,3)3																	
	11.89	FA3G3S(3,3,6)3																	
	11.89	**FA3G3S(3,6,6)3**																	
	11.89	FA3G3S(6,6,6)3																	
GP41	12.03	* A3G3S(3,3,3)3																	
	12.03	* A3G3S(3,3,6)3																	
	12.03	* A3G3S(3,6,6)3																	
	12.03	**A3F1G3S(3,3,3)3**																	
	12.03	A3F1G3S(3,3,6)3																	
	12.03	FA3BG3S(3,3,3)3																	
	12.03	FA3BG3S(6,6,6)3																	
GP42	12.15	**A4G4S(3,3,3)3**																	
	12.15	A3F1G3S(3,6,6)3																	
GP43	12.33	**A4G4S(3,3,6)3**																	
	12.33	**A4G4S(3,6,6)3**																	
GP44	12.48	**A4F1G3S(3,3,3)3**																	
	12.48	A3F2G3S(3,3,3)3																	
	12.48	**A4F1G3S(3,3,6)3**																	
	12.48	**A4F1G3S(3,6,6)3**																	
GP45	12.67	A3F2G3S(3,3,6)3																	**33%**
	12.67	A4F2G3S(3,3,3)3																	
	12.67	A4F2G3S(3,3,6)3																	
	12.67	* A4G4S(3,3,3,3)4																	
	12.78	**A4G4S(3,3,3,3)4**																	
GP46	12.96	**A4G4S(3,3,3,6)4**																	
GP47	13.27	*** A4G4S(3,3,3,6)4**																	
	13.27	A4G4S(3,3,6,6)4																	
	13.27	A4G4S(3,6,6,6)4																	
GP48	13.47	*** A4G4S(3,3,3,3)4**																**50%**	
	13.47	FA4G4S(3,3,3,3)4																	
	13.47	FA4G4S(3,3,3,6)4																	
	13.47	A4BG4S(3,3,6,6)4																	
GP49	13.82	A4F1G4S(3,3,3,3)4																	
	13.82	**A4F1G4S(3,3,3,6)4**																	
	13.82	A4F1G4S(3,3,6,6)4																	
	13.82	A4F1G4S(3,6,6,6)4																	
GP50	13.99	A4G4LacS(3,3,3,3)4																	**33%**
	13.99	A4G4LacS(3,3,3,6)4																	
	13.99	A4F2G4S(3,3,3,3)4																	
	13.99	A4F2G4S(3,3,6,6)4																	
	14.43	**A4F3G4S(3,3,3,3)4**																	

The HILIC-chromatogram was separated into 50 peaks and structural assignment carried out according to Saldova et al., 2014 [10]. Major glycans in each peak are in bold. * Sialic acids isomers (same composition but different sialic acid linkage arrangements resulting in different glucose units (GUs) from the original structures). ^#^ Peaks calculated into specific features are highlighted in grey (where there is 33 or 50%; the glycans with the given feature are approximately that abundant in the given peak): sialylation: S0 (neutral, GP1–11 + (GP12/2) + 14 + 16 + 17 + 24); S1 (monosialylated, (GP12/2) + GP13 + GP15 + GP18−23 + GP25 + GP27 + GP34); S2 (disialylated, GP26 + 28−33 + GP35); S3 (trisialylated, GP36−44); S4 (tetrasialylated, GP45−50); galactosylation: G0 (agalactosylated, GP1−2 + GP4−5 + (GP6/2) + (GP12/2)); G1 (monogalactosylated, GP3 + GP7−10 + (GP12/2) + GP13 + GP15 + GP18−19 + (GP22/2)); G2 (digalactosylated, GP14 + GP16−17 + GP20−21 + (GP22/2) + GP23 + GP25−31); G3 (trigalactosylated, GP32–33 + GP35−41); G4 (tetragalactosylated, GP34 + GP42−50); branching: A1 (monoantennary, GP1−3 + (GP12/2) + 13 + (GP15/2) + (GP22/2)); A2 (biantennary, GP4–5 + (GP6/2) + GP7–10 + (GP12/2) + GP14 + (GP15/2) + GP16–21 + (GP22/2) + GP23 + GP25−31); A3 (triantennary, GP32–33 + GP35−41); A4 (tetraantennary, GP34 + GP42−50); oligomannose: (GP6/2) + GP11; fucosylation: core-fucose: GP2 + GP5 + (GP6/2) + GP8−10 + (GP15/2) + GP16–17 + GP19 + GP23 + GP25 + GP30–31 + GP40 + (GP48/2); outer-arm fucose: GP41 + GP44 + (GP45/3) + GP49 + (GP50/3). Structure abbreviations: all *N*-glycans have two core GlcNAcs; F at the start of the abbreviation indicates a core-fucose α1,6-linked to the inner GlcNAc; Mx, number (x) of mannose on core GlcNAcs; D1 indicates that the α1-2 mannose is on the Manα1-6Manα1-6 arm, D2 on the Manα1-3Manα1-6 arm, D3 on the Manα1-3 arm of M6 and on the Manα1-2Manα1-3 arm of M7 and M8; Ax, number of antenna (GlcNAc) on trimannosyl core; A2, biantennary with both GlcNAcs as β1,2-linked; A3, triantennary with a GlcNAc linked β1,2 to both mannose and the third GlcNAc linked β1,4 to the α1,3 linked mannose; A4, GlcNAcs linked as A3 with additional GlcNAc β1,6 linked to α1,6 mannose; B, bisecting GlcNAc linked β1,4 to β1,3 mannose; Gx, number (x) of β1,4 linked galactose on antenna; F(x), number (x) of fucose linked α1,3 to antenna GlcNAc; Sx, number (x) of sialic acids linked to galactose; Lac(x), number (x) of lactosamine (Galβ1-4GlcNAc) extensions.

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
