# Peer review of "Glycosylation in Indolent, Significant and Aggressive Prostate Cancer by Automated High-Throughput N-Glycan Profiling"

_ijms, 2020, doi:10.3390/ijms21239233_

Round 1
Reviewer 1 Report
The ms is sounding good and properly articulated the results and experiments. The introduction should be improved, the authors only discussed about USA, how about world wide cases and mortality.
Rewrite the entire introduction part by addressing the PCa worldwide and incorporate references.
Author Response
Dear Sir/Madam,
thank you for your suggestions.
PCa worldwide cases and mortality replaced only USA figures including the reference in the Introduction.
Kind Regards,
Radka Saldova et al.
Reviewer 2 Report
In the manuscript "Glycosylation in indolent, significant and aggressive prostate cancer by automated high-throughput N-3 glycan profiling," authors using ultra-performance liquid chromatography analyzed the whole serum N-glycan profiling from 117 prostate cancer patients. Methodologically, this story was done very well, and there are no doubts about the authors' professionalism as glycobiologists and data acquisition. However, from the clinical point, the manuscript lacks a critical piece of information to conclude whether these results can be used for the prognosis of prostate cancer progression.
- It is unclear what part of patients with indolent and significant stages underwent androgen-deprivation therapy (ADT). Given that ADT can block the progression of disease at least for a while, it may affect these patients' glycan profile. In other words, the lack of difference, or, conversely, the detected difference between the indolent and significant groups of patients, could be attributed to this aspect.
- These data directly contradict with publications of others. It is generally accepted that prostate cancer advancement (at least at the tissue value) correlates with the downregulation of MGAT3-mediated glycosylation and the formation of bisecting structure. In the meantime, the contribution of MGAT5 is increased, and the production of tri- and tetraantennary structures are enhanced. Indeed, bisecting GlcNAc blocks branching; however, this is not the case for the data presented here. For instance, the pick of bisecting structure A2[3]G1S[3]1 demonstrates the substantial difference between significant and aggressive groups; however, when it comes to the A4G4S[3]1 tetraantennary glycans, this was not detected. Authors should take that into account, making conclusions, and claiming the potential clinical implications. Also, the literature's discrepancy between tissue and serum glycan profiles should be discussed. Otherwise, these data have only descriptive value.
- The authors mentioned that they detected the increases in hybrid and oligomannosylated structures (M5A1G1S1 and M6); however, these data are not presented.
- Minor corrections: Line 196: Peak 22 is a complex type structure, not a hybrid.
- The indication of p-value in figures could be better to facilitate readers' understanding.
Author Response
Dear Sir/Madam,
Thank you for your suggestions.
Our specific point-by-point responses are found below and we have numerated your queries to assist in the review process.
1. It is unclear what part of patients with indolent and significant stages underwent androgen-deprivation therapy (ADT). Given that ADT can block the progression of disease at least for a while, it may affect these patients' glycan profile. In other words, the lack of difference, or, conversely, the detected difference between the indolent and significant groups of patients, could be attributed to this aspect.
Response 1. None of the patients received ADT therapy as it was a radical prostatectomy cohort and we were predicting the pathology of their disease pre-operatively. The patients were put into the three groups (indolent, significant and aggressive) based on the pathology of their disease after surgery. This was clarified in the paper.
2. These data directly contradict with publications of others. It is generally accepted that prostate cancer advancement (at least at the tissue value) correlates with the downregulation of MGAT3-mediated glycosylation and the formation of bisecting structure. In the meantime, the contribution of MGAT5 is increased, and the production of tri- and tetraantennary structures are enhanced. Indeed, bisecting GlcNAc blocks branching; however, this is not the case for the data presented here. For instance, the pick of bisecting structure A2[3]G1S[3]1 demonstrates the substantial difference between significant and aggressive groups; however, when it comes to the A4G4S[3]1 tetraantennary glycans, this was not detected. Authors should take that into account, making conclusions, and claiming the potential clinical implications. Also, the literature's discrepancy between tissue and serum glycan profiles should be discussed. Otherwise, these data have only descriptive value.
Response 2. Literature discrepancy between tissue and serum glycan profiles was added to the Discussion. Conclusions took in account these discrepancies.
3. The authors mentioned that they detected the increases in hybrid and oligomannosylated structures (M5A1G1S1 and M6); however, these data are not presented.
Response 3. These named hybrid and oligomannosylated glycans (M5A1G1S1 and M6) are about the same abundancy together as the most abundant glycan A2G2S1 in the peak 22. This was clarified.
4. Minor corrections: Line 196: Peak 22 is a complex type structure, not a hybrid.
Response 4. This was clarified as explained above.
5. The indication of p-value in figures could be better to facilitate readers' understanding.
Response 5. The p-values were indicated better in the figures 2 and 3.
Kind Regards,
Radka Saldova et al.